The association between urbanization and adolescent depression in China

Pan Degong 1
Yan Ning 2
Pu Lining 1
He Xiaoxue 1
Wang Huihui 1
Zhang Xue 1
Shi Xiaojuan 1
Wen Jing 1
Li Jiangping lijp@nxmu.edu.cn 1 3
1 Department of Epidemiology and Health Statistics, School of Public Health, Ningxia Medical University , Yinchuan , China
2 Heart Centre & Department of Cardiovascular Diseases, General Hospital of Ningxia Medical University , Yinchuan , China
3 Key Laboratory of Environmental Factors and Chronic Disease Control, Ningxia Medical University , Yinchuan , China
Chen Chong
Electronic publication date: 2024 Feb 22
Publication date: 2024
Volume: 12
Electronic Location ID: e16888
Received 2023 Aug 11; Accepted 2024 Jan 15
Copyright: ©2024 Pan et al.
Copyright year: 2024
Copyright holder: Pan et al.
License: This is an open access article distributed under the terms of the Creative Commons Attribution License, which permits unrestricted use, distribution, reproduction and adaptation in any medium and for any purpose provided that it is properly attributed. For attribution, the original author(s), title, publication source (PeerJ) and either DOI or URL of the article must be cited.
License URL: https://creativecommons.org/licenses/by/4.0/

Keywords: Adolescent, China, Depression, Urbanization

Funding: Natural Science Foundation of Ningxia 2023AAC03213 This research was supported by the Natural Science Foundation of Ningxia (2023AAC03213). The funders had no role in study design, data collection and analysis, decision to publish, or preparation of the manuscript.

==============================
Background

With the rapid urbanization in many countries, more attention is being paid to the relationship between urbanization and mental health, especially depression. However, in countries with rapid urbanization, few empirical studies exist on the relationship between urbanization and adolescent depression.

Methods

Nationally representative survey data from the China Family Panel Studies in 2012, 2016 and 2018 were used. Data of 1,588 adolescents were obtained from 25 provinces. Depression was measured using the Center for Epidemiology Studies of Depression 20-item score. The urbanization rate was obtained from the National Bureau of Statistics of China. The generalized estimating equation was used to estimate the statistical relationship.

Results

The participants’ mean age at baseline was 15 years, and 51.2% (813/1,588) of participants were male. After adjusting for all covariates (gender, age, ethnicity, level of education, marital status, urban/rural areas, body mass index, self-rated health, academic pressure, smoking, drinking and exercise), the rate of urbanization was monotonically and negatively associated with adolescent depression (odds ratio 0.34, 95% CI [0.14–0.79]). Compared with female adolescents, male adolescents had a lower risk of depression (odds ratio 0.80, 95% CI [0.67–0.97]).

Conclusion

In the context of China, urbanization has a positive effect on the mental health of adolescents. Female adolescents are more likely to experience depression than male adolescents.

Introduction

In the 21st century, urbanization represents a major demographic shift in developed and developing countries (Zuo et al., 2018). China is experiencing an unprecedented process of urbanization. China’s urban population growth has the characteristics of rural–urban migration (Gong et al., 2012). It is estimated that more than one billion people will live in cities in China by 2030 (Wang et al., 2018). Although urbanization provides people with education, medical, and other conveniences, it is associated with negative effects such as environmental pollution (Sun, Wang & Zhu, 2023; Yue et al., 2020). Therefore, rapid urbanization has a profound and lasting impact on local and even national public health (Li et al., 2016b). The impact of urbanization on individual mental health, especially on depression, has attracted considerable academic interest.

Depression is a common mental disorder. It is a state of mind characterized by persistent low mood, loss of enjoyment in daily activities, and loss of energy, leading to varying degrees of impaired social and occupational functioning that can severely reduce quality of life (Lim et al., 2018; Malhi & Mann, 2018). Depression is common, expensive to treat and is associated with medication side effects and an increased risk of suicide (Marwaha et al., 2023). In recent years, adolescent depression poses a public health concern. Depression often first develops during adolescence, and the incidence of depression rises sharply after adolescence (Izaki, 2021). A recent epidemiological study reported that the 12-month prevalence of depressive disorders in adolescents was 8.2%, and the prevalence in girls was about twice that in boys (Kessler et al., 2012). In the past few years, the global trend is for increases in adolescent depression (Shorey, Ng & Wong, 2022). A meta-analysis has suggested that depressive symptoms are becoming more prevalent among Chinese adolescents (Li et al., 2019). Moreover, Chinese researchers have found that rural areas are still a potential risk factor for the development of depressive traits in Chinese adolescent students (Chen, Huang & Riad, 2021). The worst possible outcome of depression is suicide, which is the second leading cause of death in adolescents. Adolescent depression, if not controlled, can lead to physical and mental health problems in later life (Hestetun, Svendsen & Oellingrath, 2015). Thus, it is essential to pay urgent attention to the mental health of adolescents.

Urbanization plays an important role in the mental health of individuals (Hong et al., 2022). Previous studies have shown that most rural environments are better for mental health than urban environments, and polluted environments are often detrimental to the mental health of residents. As a result, common psychiatric syndromes, including depression, are more prevalent in urban areas (Gruebner et al., 2017; Ventriglio et al., 2021). At the same time, owing to the fast pace of life, people living in cities are more vulnerable to personal and social pressures, which increase the likelihood of depression (Lecic-Tosevski, 2019; Richaud & Amin, 2019). Although urban residence is considered to be a risk factor for depression and other mental disorders, epidemiological evidence is inconsistent. Of note, the prevalence of major depression and other severe mental disorders is not higher in most urban areas compared with most rural areas (Breslau et al., 2014). Studies have also shown that planned urbanization reduces depressive symptoms (Hong et al., 2022), which suggests that it is too early to interpret the association between urbanization and mental health.

Many studies have shown a relationship between urbanization and mental health in China. However, most were cross-sectional studies or involved middle-aged or elderly participants. Whether there is a similar relationship between depression and urbanization has not been determined in adolescents. Compared with horizontal research, longitudinal research involves the analysis of long-term data, to enable a deep exploration of the relationship between variables. Therefore, this study examined longitudinal associations between depression and urbanization in the provinces in which adolescents lived at three different time points. Identifying the risk factors responsible for adolescent depression and determining the association between depression and urbanization would pave path for future research and also helps stakeholders to create policies to address such issues.

Methods

Study participants

Data were obtained from the China Family Panel Studies (CFPS, funded by Peking University) and carried out by the Institute of Social Science Research of Peking University (Xie & Lu, 2015). The CFPS is an ongoing nationally representative longitudinal household survey covering 95% of the population in 25 provinces in China. The sampling method is based on the multi-stage approach, and the target sample size of the CFPS was 16,000 households (Xie & Lu, 2015; Zhou, Fan & Yin, 2018). It aims to collect data from families and individuals on a range of topics, including educational background, work status, physical and mental health every 2 years. Thus, the population in this study had a high level of representativeness. The survey activities used telephone follow-up, recording monitoring, computer operation playback and other multi-clock methods for quality control. The data quality of the survey sample was also affirmed by the academic community (Zhou, Fan & Yin, 2018).

The CFPS officially started with a baseline survey in 2010. Since the CFPS did not have data on the 2010 and 2014 measures of adolescent depression, we determined to use the 2012, 2016 and 2018 measures of adolescent depression for the study. The adolescents between the age group of 10–24 years, willing to participate in the study, and present at each three rounds of survey were included in the study (Sawyer et al., 2018). Prior to being surveyed, each participant in the study signed a written informed consent form. At the same time, parents of younger/minor study participants consented to participate in the survey. A total of 1,588 adolescents were included after excluding participants with missing covariate data (Fig. 1).

Figure 1 Flow chart of the study population.

Measurement of depression

The dependent variable was depression, which was assessed using the CES-D 20-item score (Center for Epidemiology Studies of Depression 20-item). Which has been shown to be applicable to adolescents (Radloff, 1991). The CES-D 20-item score includes 20 questions to measure the mental state of residents within a week. These include the experience of feeling lonely or disliked and the experience of people being unfriendly (Mulvaney & Kendrick, 2005). The initial subjects for this study in 2012 were obtained from two databases, the CFPS 2012 child and CFPS 2012 adult. The Cronbach’s alpha for the CES-D 20-item of them were 0.8092 and 0.8486, respectively. Which is regarded as satisfactory and acceptable (Taber, 2018).

It required participants to rate the frequency of each symptom in the past week from 1 (“little or no [<1 day]”) to 4 (“most of the time [5–7 days]”). The total score ranged from 20 to 80 points; higher scores indicated higher levels of depression. Previous studies have proposed a ≥36 cut-off for depression (Radloff, 1977). Therefore, depression was dichotomized in this study. CES-D scores higher than 35 were defined as depression, whereas scores ≤35 were defined as no depression.

Measurement of urbanization

Urbanization refers to the transfer of population from rural to urban areas. The National Bureau of Statistics of China (NBSC) defines urban areas as the combination of city district areas and township areas. City districts means urban built-up areas where the seats of the municipal/district government are situated and their connective urban built-up areas. Townships means urban built-up areas where the seats of the town government are situated and their connective urban built-up areas. Urban residents are defined as residents who have lived in urban areas for six consecutive months (Wang et al., 2018). Urbanization rate is used as a proxy for urbanization in this study. This study measured the urbanization rate as the ratio of usual urban residents to total population at the prefecture level. By using NBSC statistical data of the urbanization rate of each province, we obtained the urbanization rate of the target provinces including Beijing in the three years of the study (2012, 2016 and 2018).

Control variables

We adjusted sociodemographic characteristics, health status, and lifestyle related items at baseline in the model. The sociodemographic characteristics were gender, age, ethnicity (Han/minorities), marital status (married/single), level of education (primary school or less/junior high/high school/college or more), and urban/rural areas (urban areas/rural areas). Health status included self-rated health, academic pressure, and body mass index (BMI). The individual education variable is the highest degree earned during the 2012, 2016, and 2018 surveys. Academic pressure was measured by the yes/no question “How’s your academic pressure?” in 2012 survey. The BMI (kg/m2) was divided into four groups: underweight (<18.5); normal (18.5–23.9); overweight (24.0–27.9); and obesity (>28). The self-rated health was rated on a 5-point scale of “excellent”, “ very good”, “good”, “fair”, and “poor”. Three lifestyle-related items were considered in the present study: smoking, which was evaluated by the yes/no question “Have you smoked cigarettes in the last month?”; drinking was measured by the yes/no question “In the past month, did you drink alcohol 3 times a week?” (Pan et al., 2022); and exercise, which was evaluated by the question “What is the frequency of exercise?”.

Statistical analysis

We exported data from the NBSC and CFPS databases for analysis. Descriptive analyses were used to produce means (standard deviations) for continuous variables and numbers (percentages) for categorical variables.

The generalized estimating equations (GEE) uses quasi-likelihood estimation to extend the general linear model to longitudinal data, estimating changes over time at the overall level rather than at the individual or specific cluster level (Bui, Vu & Tran, 2018). Therefore, the GEE model was designed to explore crude and adjusted associations between urbanization and adolescent depression. Model 1 only included the dependent variable (depression) and the urbanization. Model 2 included sociodemographic characteristics added to model 1 (include gender, age, urban/rural areas, ethnicity, marital status, and level of education). Model 3 added health status covariates to model 2 (BMI, academic pressure and self-rated health). Lastly, model 4 was a fully adjusted model with lifestyle-related items (smoking, drinking and exercise) added to model 3, it was a fully adjusted model. The GEE model was used to examine whether the association of urbanization and adolescent depression differed by sex.

Statistical significance was set at p < 0.05. Analysis was performed using the Stata MP 16.0 software package (Stata Corp LP, College Station, TX, USA).

Results

Participant characteristics

Through three follow-up surveys, a total of 1,588 adolescents were included in this study. The GEE models included invariant variables and time-varying variables. The invariant variables were the respondents’ gender, age and academic pressure. All the other variables were time-varying variables. The characteristics of participants by cohort are shown in Table 1. The mean (standard deviation) age was 15.084 (3.524) years at baseline, and 51.20% of participants (813/1,588) were male. Approximately 90% of the participants were Han. The number of participants living in the city increased significantly from 698 in 2012 to 885 in 2018. There were 19.46% participants with depression at baseline, and 13.66% and 19.02% with depression at the second and third surveys, respectively. The average urbanization rate showed an upward trend from 51.9% in 2012 to 59.4% in 2018.

Table 1 Characteristics of 1,588 Chinese adolescents at baseline by cohort.

		2012	2016	2018	
		n	%	n	%	n	%	
Depression								
	No	1,279	80.54	1,371	86.34	1,286	80.98	
	Yes	309	19.46	217	13.66	302	19.02	
Urbanization rate	mean (SD)	0.519	(0.133)	0.567	(0.114)	0.594	(0.107)	
Age	mean (SD)	15.084	(3.524)	19.088	(3.521)	21.089	(3.518)	
Gender								
	Female	775	48.80	–	–	
	Male	813	51.20	–	–	
Ethnicity code							
	Minorities	154	9.70	–	–	
	Han	1,434	90.30	–	–	
Academic pressure							
	No	440	27.71	–	–	
	Yes	1,148	72.29	–	–	
Level of education								
	Primary school or less	1,002	63.10	373	23.49	83	5.23	
	Junior high	394	24.81	584	36.78	560	35.26	
	High school	173	10.89	377	23.74	614	38.66	
	College or more	19	1.20	254	15.99	331	20.84	
Marital status							
	Single	1,520	95.72	1,414	89.04	1,325	83.44	
	Married	68	4.28	174	10.96	263	16.56	
Urban/rural areas							
	Rural areas	890	56.05	786	49.50	703	44.27	
	Urban areas	698	43.95	802	50.50	885	55.73	
BMI								
	<18.5	753	47.42	455	28.65	387	24.37	
	18.5–24.0	683	43.01	921	58.00	923	58.12	
	24.0–28.0	115	7.24	159	10.01	202	12.72	
	≥28.0	37	2.33	53	3.34	76	4.79	
Self-rated health							
	poor health	19	1.20	21	1.32	35	2.20	
	fair	92	5.79	115	7.24	57	3.59	
	good	478	30.10	507	31.93	649	40.87	
	very good	607	38.22	547	34.45	495	31.17	
	excellent	392	24.69	398	25.06	352	22.17	
Smoking							
	No	1,498	94.33	1,401	88.22	1,335	84.07	
	Yes	90	5.67	187	11.78	253	15.93	
Drinking								
	No	1,548	97.48	1,474	92.82	1,503	96.45	
	Yes	40	2.52	114	7.18	85	5.35	
Exercise								
	No	267	16.81	688	43.32	639	40.24	
	Yes	1,321	83.19	900	56.68	949	59.76	

Association of depression with urbanization

Table 2 shows the association of depression with urbanization. In model 1, which included only the urbanization variable, urbanization negatively influenced depression. This indicates a significant negative relationship between urbanization and respondents’ depression. As shown in model 2, urbanization, ethnicity and gender were significantly associated with depression in adolescents. When variables related to health status were added to model 2, the correlation between ethnicity and depression was not significant in model 3. Depression was negatively influenced by urbanization, male adolescents, and self-rated health. After adding lifestyle-related items to model 3 in model 4, urbanization, male adolescents, and self-rated health were still negatively correlated with depression.

Table 2 Relationship between covariates and depression in GEE.

	Model1	Model2	Model3	Model4	
Variables	OR (95%CI)	P	OR (95%CI)	P	OR (95%CI)	P	OR (95%CI)	P	
Urbanization	0.31 [0.14,0.67]	0.003**	0.38 [0.17,0.87]	0.023*	0.36 [0.16,0.83]	0.017*	0.34 [0.14,0.79]	0.012*	
Gender			0.80 [0.67,0.96]	0.017*	0.83 [0.69,0.99]	0.038*	0.80 [0.67,0.97]	0.020*	
Age			1.01 [0.99, 1.04]	0.368	1.02 [0.99,1.05]	0.287	1.02 [0.98,1.05]	0.283	
Ethnicity code			0.74 [0.57,0.96]	0.023*	0.77 [0.59,1.02]	0.064	0.78 [0.59,1.03]	0.078	
Level of education			0.95 [0.85,1.07]	0.408	0.92 [0.82,1.05]	0.215	0.91 [0.80,1.03]	0.152	
Marital status			0.93 [0.78,1.11]	0.411	1.21 [0.92,1.59]	0.163	1.19 [0.90,1.57]	0.233	
Urban/rural areas			0.93 [0.78,1.11]	0.411	0.92 [0.77,1.09]	0.321	0.91 [0.77,1.08]	0.282	
BMI					0.91 [0.81,1.03]	0.138	0.91 [0.81,1.03]	0.141	
Self-rated health					0.62 [0.57,0.68]	<0.000***	0.62 [0.57,0.67]	<0.001***	
Academic pressure					1.19 [0.95,1.49]	0.129	1.14 [0.90,1.43]	0.275	
Smoking							1.09 [0.91,1.30]	0.365	
Drinking							1.86 [0.60,1.21]	0.376	
Exercise							1.25 [1.02,1.53]	0.030*	
Notes.

* P <0.05.

** P <0.01.

*** P <0.001.

Association of depression with urbanization in different genders

As shown in Table 3, the prevalence of depression in male adolescent living in regions with a high urbanization rate was lower than that in regions with a low urbanization rate. However, there was no significant difference in the prevalence of depression among female adolescents living in regions with different urbanization.

Table 3 Relationship between covariates and depression in different genders.

	Male	Female	
	OR (95%CI)	P	OR (95%CI)	P	
Urbanization	0.23 [0.07,0.77]	0.017*	0.45 [0.14,1.50]	0.196	
Age	1.01 [0.96,1.05]	0.780	1.04 [0.98,1.09]	0.199	
Ethnicity code	0.89 [0.61,1.31]	0.558	0.69 [0.47,1.02]	0.066	
Level of education	1.00 [0.83,1.19]	0.968	0.84 [0.70,1.01]	0.062	
Marital status	1.08 [0.76,1.55]	0.654	1.36 [0.81,2.27]	0.243	
Urban/rural areas	0.95 [0.73,1.23]	0.695	0.88 [0.70, 1.11]	0.287	
BMI	0.90 [0.76,1.06]	0.195	0.91 [0.75,1.10]	0.320	
Self-rated health	0.63 [0.55,0.70]	<0.001***	0.61 [0.53,0.70]	<0.001***	
Academic pressure	1.00 [0.73,1.37]	1.000	1.29 [0.91,1.83]	0.148	
Smoking	0.96 [0.75,1.24]	0.774	1.22 [0.94,1.59]	0.128	
Drinking	0.91 [0.57,1.44]	0.684	0.84 [0.46,1.53]	0.565	
Exercise	1.22 [0.93,1.62]	0.154	1.25 [0.93,1.68]	0.142	
Notes.

* P <0.05.

** P <0.01.

*** P <0.001.

Discussion

The purpose of this study was to explore the relationship between urbanization and adolescent depression. In this national longitudinal program study, after adjusting for sociodemographic characteristics, health status and lifestyle-related items, we found that the urbanization was monotonically and negatively associated with depression. Participants with poor health status may have a higher prevalence of depression during the subsequent 6 years. This study also showed that female adolescents were more likely to suffer from depression than male adolescents.

Many studies have shown that the process of urbanization tends to increase the burden of mental ill health (Cortina & Hardin, 2023; Ventriglio et al., 2021). To the contrast, the findings of this study suggested that depression is less prevalent in more urbanized areas in China. Firstly, some rural regions lack sufficient medical and health resources to treat depression and related mental health conditions. For example, in 2017–18, about one-third of urban counties had no psychiatrists per 100,000 residents, compared with 79 percent in adjacent rural areas (Kirby et al., 2019). Furthermore, there are many barriers to care in rural areas, including lack of infrastructure and substance abuse (Thomas, Macdowell & Glasser, 2012). Secondly, compared with residents living in areas with high urbanization, those living in areas with lower urbanization have less awareness of depression and its severity, as rural residents may have limited access to sources of health information (Chen et al., 2019). Lack of awareness of depression and its symptoms could affect rural residents willingness to receive diagnosis or treatment, resulting in low utilization of basic public health services (Jadnanansing et al., 2022; Li et al., 2011; Phillips et al., 2009). Thus, urbanization can provide opportunities to protect health, especially for people with depression, such as better living conditions, dissemination of better health knowledge, and improved access to health care (Purtle et al., 2019; van der Wal et al., 2021; Yang et al., 2013).

In line with the previous study (Hankin & Abramson, 1999; Seiffge-Krenke, 2007), this study also showed a gender difference, with female adolescents having worse mental well-being than male adolescents. One study found that when young women are confronted with negative life events, they are more inclined to attribute the cause to their own lack of ability or personality faults, so under more psychological pressure (Parker & Brotchie, 2010). We also found that the prevalence of depression among male adolescents in areas with higher urbanization was lower than that in areas with low urbanization. However, there was no significant difference in depression between areas of low and high urbanization among female adolescents. Before the 21st century, girls were a high-risk group for mental ill health in rural areas because of severe gender inequality in rural China, and suicidal behavior and thoughts were more prevalent in girls from rural than urban areas (Chen, Huang & Riad, 2021). However, with the implementation of gender equality policies in recent decades, the difference of female mental health between urban and rural areas may decrease. In the past, rural women had very few opportunities to receive education, and depression was associated with lower educational attainment (Wickersham et al., 2021). In Chinese cities in the 1990s, women were encouraged to be independent and pursue higher education by centralized policies (Greenberger et al., 2000; Hershatter & Zheng, 2008). For example, the Chinese government introduced a rural teacher support program in 2015 to improve the quality of education for every rural child and adolescent (Chen, Huang & Riad, 2021). These political efforts have partially eliminated differences in the mental health of girls between urban and rural areas.

In developing countries, more attention is being paid to the relationship between urbanization and health. Urbanization may improve some health problems faced by developing countries and may also worsen other health problems (Eckert & Kohler, 2014). This study shows that urbanization has a positive impact on adolescent mental health. However, in the process of China’s urbanization, there is a need for innovative health policies that continue to meet the health-care needs of new urban residents while further upgrading the level of health-care services for rural residents. Female adolescents from China have a higher risk of developing depression. Therefore, researchers should develop more gender-specific and education-related intervention programs. In addition to policies to promote aggressive treatment for existing adolescents with depression, policies should further focus on supportive palliative measures to curb depressive symptoms in adolescents before they escalate (Shorey, Ng & Wong, 2022). Attention to the mental health of adolescents should be strengthened, especially for female adolescents, to improve their understanding of mental health and enhance their intention to receive treatment for mental health problems. This will involve improving the availability of adolescent mental health services, increasing mental health literacy, and promoting help-seeking behaviors for mental health difficulties.

The strengths of this study included complete panel data, its prospective nature, and a community-based representative sample. Secondly, to our knowledge, previous studies have focused on the relationship between depression among adults and older adults in the context of urbanization (Hu, Li & Martikainen, 2019; Li et al., 2016a; Li et al., 2011), whereas few studies have investigated the relationship between depression and urbanization in a national cohort of Chinese adolescents. However, this study had two limitations. Firstly, participants from some provinces in China were not included in the three cohort follow-up visits. Secondly, the variables controlled in this study were mainly individual factors and we did not include potential influencing factors at the community or society level.

Conclusion

This study investigated the relationship between urbanization and depression of adolescents in China using GEE. Specifically, urbanization was negatively correlated with adolescent depression. The main reason for this discrepancy may be due to the medical resources gap between regions with different levels of urbanization. In summary, with rapid urbanization, we should pay more attention to the mental health of adolescents, especially depression.

Supplemental Information

Supplemental Information 1 Source data for statistical description and correlation analysis in manuscripts

Supplemental Information 2 Process data commands

Supplemental Information 3 Description of the data classification

The authors sincerely thank the China Family Panel Studies (CFPS) for providing consent for data access, and the Institute of Social Science Survey of Peking University for assisting us in conducting this study. We also want to give a special acknowledgement to the National Bureau of Statistics of China (NBSC) for providing real and reliable data for academic research.

Additional Information and Declarations

Competing Interests

Author Contributions

Data Availability

The authors declare there are no competing interests.

Degong Pan conceived and designed the experiments, performed the experiments, analyzed the data, prepared figures and/or tables, authored or reviewed drafts of the article, and approved the final draft.

Ning Yan conceived and designed the experiments, performed the experiments, analyzed the data, authored or reviewed drafts of the article, and approved the final draft.

Lining Pu conceived and designed the experiments, analyzed the data, prepared figures and/or tables, and approved the final draft.

Xiaoxue He conceived and designed the experiments, performed the experiments, prepared figures and/or tables, and approved the final draft.

Huihui Wang performed the experiments, analyzed the data, authored or reviewed drafts of the article, and approved the final draft.

Xue Zhang performed the experiments, analyzed the data, authored or reviewed drafts of the article, and approved the final draft.

Xiaojuan Shi performed the experiments, analyzed the data, prepared figures and/or tables, authored or reviewed drafts of the article, and approved the final draft.

Jing Wen performed the experiments, prepared figures and/or tables, authored or reviewed drafts of the article, and approved the final draft.

Jiangping Li conceived and designed the experiments, performed the experiments, analyzed the data, authored or reviewed drafts of the article, and approved the final draft.

The following information was supplied regarding data availability:

The data is available at China Family Panel Studies (CFPS), a social survey sponsored by the Institute of Social Science Survey (ISSS) of Peking University CFPS: CFPS Public Data: CFPS 2012, CFPS 2016 and CFPS2018, http://www.isss.pku.edu.cn/cfps.

The raw measurements are available in the Supplemental Files.

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
