# Peer review of "The association between urbanization and adolescent depression in China"

_PeerJ, doi:10.7717/peerj.16888_

## Round 0.1 · original submission · Major Revisions

Please address multiple methodological issues raised by the reviewers.

**Language Note:** The review process has identified that the English language must be improved. PeerJ can provide language editing services - please contact us at [email protected] for pricing (be sure to provide your manuscript number and title). Alternatively, you should make your own arrangements to improve the language quality and provide details in your response letter. – PeerJ Staff

·

Basic reporting

(1)The English language should be improved to ensure that an international audience can clearly understand your text.Some examples where the language could be improved include lines 104, 105, 129, 130 the current phrasing makes comprehension difficult. I suggest you have a colleague who is proficient in English and familiar with the subject matter review your manuscript, or contact a professional editing service.
(2)Depression as a key concept of the manuscript, should be more detailed in the introduction and also in relation to urbanization.
(3)You decide to refer to the subjects aged 10 to 24 as adolescents, considering that they ranged from 10 to 24, and your three data participants' average ages were 15, 19, and 21, respectively, covering the early, middle, and late stages of adolescence, which have significant developmental differences. During adolescence, early adolescents are particularly vulnerable to stress (Holder & Blaustein. 2018), which may be different from middle and late adolescents. Therefore, how to determine whether it is the changes in depression brought about by urbanization rather than the different stage characteristics of adolescents?

Experimental design

The dependent variable is depression, but your study only uses the detection rate of depression to represent depression and lacks indicators such as the severity of depression, which needs to be clarified.

Validity of the findings

Lines 253-255 of the manuscript are not part of the conclusion and need to be adjusted.

Additional comments

Please add some practical implications for the findings of this paper.

Reviewer 2 ·

Basic reporting

Authors presented an interesting study testing the hypothesis that urbanisation affect mental health (depression) in adolescents. Authors gather data from two nation databases in order to test their hypothesis using GEE models.
The manuscript is well written with minor suggestions for editing as presented bellow:

Minor: In the abstract, line 47, authors wrote: “The participants average age is about”. Please, change it to past tense.
Minor: in the abstract, line 48, authors wrote: “all covariates”, please, if the space allows, include the list of covariates.
Minor: I would suggest change the subtitle for Methods and Data

Experimental design

Yes, the manuscript meet the main standards for PeerJ. Research question is clear and well defined, as well as the methods are appropriate to test their hypothesis. However, I made several notes that I would like authors to address in order to improve their manuscript.

Abstract:

Major: authors mentioned at line 50 that females had higher risk of depression. This affirmation is not supported by the data presented on the results section. Please, double-check if it is not inverted (males had higher risk of depression than females).

Methods:
Question: authors mentioned that CFPS is collected every two years, however, they presented data from 2012, 2016 and 2018. Why is 2014 missing? Is there any reason for not considering 2014 for this analaysis?

At line 111: I believe authors wanted to justify the use of data from subjects with 24 years old. However, they might consider rewriting in a more directly manner.
At line 148: authors wrote: “exercise, which was evaluated by at least once/no question “What is the frequency of exercise”? I might be misunderstanding, but it is not clear to me if the participants had more options to answer to this question. Can authors clarify?

Validity of the findings

Please, check the following comments:

Results:
When presenting participants characteristics authors could explore how the number of subjects in rural/urban areas changed according to time. I believe this is an important factor of this study and having a figure showing the increasing number of people living in urban areas would be beneficial for the manuscript.

Something I did not understand. Authors considered depression as the dependent variable for their model, however, it is not clear to me if the considered the CES-D score for the three-study year (2012, 2016, 2018), or only the CES-D score for the last year? Also, it is not clear to me if they considered the score of CES-D as the dependent variable or being/not being over >35 points (depression) as a binary dependent variable?

When presenting results of Table 3 authors could briefly introduce the analysis. It is not clear what type of regression models were applied here.

Discussion:
I do not agree with the first paragraph of discussion. As far as I understood, females’ adolescents did not present any significant association with depression, as presented at Table 3 (OR 0.50 (0.15-1.61), p = 0.245). Furthermore, it is not clear if self-rated health is positively or negatively associated with depression, since, as far as I understood, Self-rated health was a Likert scale where higher values indicated better health. I would advise authors to revise their discussion section considering the warnings.

Conclusion:

Authors wrote at line 252: “In summary, regular urbanization is beneficial to adolescent mental health”. I would like authors to explain this. Their findings indicate that urbanization is positively associated with depression. How could urbanization be beneficial to adolescents mental health?

Additional comments

Thank you for the opportunity of reviewing this manuscript.

Reviewer 3 ·

Basic reporting

# The article has several grammatical mistakes, which needs to be corrected.
# The given details are needed to be supported by recent references.
# While drafting the article, scientific terminology is recommended to use.
Kindly see the attached pdf for comments.

Experimental design

#Details are required about study setting, sampling and recruitment. As data were gathered at three different point of time, detailed information of participants should be mentioned such as what about lost to follow up? What if someone migrated in or out in that region during data collection?
#Flow chart would be recommended to describe sampling methods.
# Validation of data collecting tool is required. Give details on quality control.
# Levels of urbanization should be defined in detail.
# How have you dealt with ethical issues? for example, what measures were taken for those who have identified with depression? was assent taken from the parents for their children to take participation in study?

Validity of the findings

# discussion and conclusion should be relevant to the study findings.
# What measures were taken to overcome the bias/limitation of the study?

Annotated reviews are not available for download in order to protect the identity of reviewers who chose to remain anonymous.

---

## Round 0.2 · Minor Revisions

Please address additional issues raised by reviewer 3.

·

Basic reporting

no comment

Experimental design

no comment

Validity of the findings

no comment

Additional comments

no comment

Reviewer 2 ·

Basic reporting

Authors did a very good job replying my previous concerns and I believe the manuscript improved significantly.
The English review also improve manuscript readability and quality.

Experimental design

I have no further comments.

Validity of the findings

I have no further comments.
Data presents important information on the relationship between Depression and urbanisation.

Reviewer 3 ·

Basic reporting

Still there is a scope for language correction and especially for the scientific writing improvements. (Go through the attached file for few corrections)

It is good that you have cited current studies in your references this time.

Experimental design

There are still some information missing even after commenting on 1st review.
For example
There is no discussion about tool validation. It should be mentioned under quality control that the tool which used for data collection was validated or not.

The study participants who were younger/minor then what about the consent of their parents?- can be mentioned in study participants.

What measures have been taken for those who have identified with depression?- can be mentioned under ethical consideration

Validity of the findings

No comments

Additional comments

Methodology is the crucial part of study design. It should be describe in such a depth that one can replicate the study without any difficulties in understanding.

Flow of the information should be maintained in methods. for example, study design; period; setting, study participants, sample size, data collection/recruitment/sampling, data collection tool, variables, quality control, statistical analysis, ethical consideration and so on.

Annotated reviews are not available for download in order to protect the identity of reviewers who chose to remain anonymous.

---

## Round 0.3 · accepted · Accept

Thank you for addressing the reviewers' concerns.

Reviewer 3 ·

Basic reporting

No Comments

Experimental design

No Comments

Validity of the findings

No Comments

Additional comments

N/A